# Prediction of Acute and Chronic Mastitis in Dairy Cows Based on Somatic Cell Score and Mid-Infrared Spectroscopy of Milk

**DOI:** 10.3390/ani12141830

**Published:** 2022-07-18

**Authors:** Lisa Rienesl, Negar Khayatzdadeh, Astrid Köck, Christa Egger-Danner, Nicolas Gengler, Clément Grelet, Laura Monica Dale, Andreas Werner, Franz-Josef Auer, Julie Leblois, Johann Sölkner

**Affiliations:** 1Department of Sustainable Agricultural Systems, University of Natural Resources and Life Sciences, 1180 Vienna, Austria; lisa.rienesl@boku.ac.at (L.R.); kh.negar@gmail.com (N.K.); 2ZuchtData EDV-Dienstleistungen GmbH, 1200 Vienna, Austria; koeck@zuchtdata.at (A.K.); egger-danner@zuchtdata.at (C.E.-D.); 3Regional Association for Performance Testing in Livestock Breeding of Baden-Wuerttemberg (LKV—Baden-Wuerttemberg), 70067 Stuttgart, Germany; nicolas.gengler@uliege.be; 4Walloon Agricultural Research Center (CRA-W), 5030 Gembloux, Belgium; c.grelet@cra.wallonie.be; 5Gembloux Agro-Bio Tech, Université de Liège (ULg), 5030 Gembloux, Belgium; ldale@lkvbw.de (L.M.D.); awerner@lkvbw.de (A.W.); 6LKV Austria Gemeinnützige GmbH, 1200 Vienna, Austria; franz.josef.auer@lkv-austria.at; 7Elevéo (Awé Groupe), 5590 Ciney, Belgium; jleblois@awegroupe.be

**Keywords:** clinical mastitis, mid-infrared (MIR) spectroscopy, somatic cell count, dairy cow, partial least squares discriminant analysis

## Abstract

**Simple Summary:**

Mid-infrared (MIR) spectroscopy is the method of choice to determine milk components like fat, protein and urea. We examined the potential of MIR spectra analyses for the prediction of clinical mastitis events of dairy cows additionally, or alternatively, to somatic cell count, which is routinely used as an indicator for mastitis monitoring. Prediction models based on MIR spectra and a somatic cell count-derived score (SCS) were developed and compared. A model based on MIR spectra and SCS proved more accurate at predicting mastitis than models based on either indicator alone. Consequently, MIR spectra analyses add extra value in the prediction of clinical mastitis, making them potentially useful for dairy farm management and as an auxiliary trait for the genetic evaluation of udder health.

**Abstract:**

Monitoring for mastitis on dairy farms is of particular importance, as it is one of the most prevalent bovine diseases. A commonly used indicator for mastitis monitoring is somatic cell count. A supplementary tool to predict mastitis risk may be mid-infrared (MIR) spectroscopy of milk. Because bovine health status can affect milk composition, this technique is already routinely used to determine standard milk components. The aim of the present study was to compare the performance of models to predict clinical mastitis based on MIR spectral data and/or somatic cell count score (SCS), and to explore differences of prediction accuracies for acute and chronic clinical mastitis diagnoses. Test-day data of the routine Austrian milk recording system and diagnosis data of its health monitoring, from 59,002 cows of the breeds Fleckvieh (dual purpose Simmental), Holstein Friesian and Brown Swiss, were used. Test-day records within 21 days before and 21 days after a mastitis diagnosis were defined as mastitis cases. Three different models (MIR, SCS, MIR + SCS) were compared, applying Partial Least Squares Discriminant Analysis. Results of external validation in the overall time window (−/+21 days) showed area under receiver operating characteristic curves (AUC) of 0.70 when based only on MIR, 0.72 when based only on SCS, and 0.76 when based on both. Considering as mastitis cases only the test-day records within 7 days after mastitis diagnosis, the corresponding areas under the curve were 0.77, 0.83 and 0.85. Hence, the model combining MIR spectral data and SCS was performing best. Mastitis probabilities derived from the prediction models are potentially valuable for routine mastitis monitoring for farmers, as well as for the genetic evaluation of the trait udder health.

## 1. Introduction

Mastitis ranks among the most prevalent diseases in dairy herds, affecting animal welfare and causing economic losses [1,2]. Hence, managing udder health and controlling mastitis are critical on dairy farms [3]. Mastitis can be clinical, presenting as inflamed udder quarters and altered milk appearance, or it can be subclinical, without clinical signs, but with an increased somatic cell count (SCC) and changes in milk composition. Within the Austrian health monitoring system (GMON), veterinarians differentiate between acute and chronic cases when diagnosing clinical mastitis [4]. Acute mastitis is characterized by sudden onset and severe signs; chronic disease, by longer persistence or recurrence [5,6].

Currently, SCC is one of the most frequently used indicators for mastitis in individual cows, as well as for monitoring prevalence of subclinical mastitis on herd level [6,7]. The SCC of an uninfected mammary gland should be less than 100,000 cells/mL according to the International Dairy Federation [8]. SCC was found to be affected by breed, parity, lactation stage, management practices and season [9].

In addition to SCC, farmers can detect mastitis events through visual observation or palpation of the udder, changes in milk secretion and application of the California Mastitis Test [5].A supplementary tool to predict mastitis may be mid-infrared (MIR) spectral analysis of milk, because bovine health status can affect milk composition. MIR spectrometry is a fast, nondestructive technology routinely used by official laboratories for quality control of milk and quantification of major milk components, such as fat, protein, casein, urea, and lactose [10]. Furthermore, it has been applied to determine additional milk components such as fatty acids [11], protein fractions [12] and minerals [13]. MIR spectrometry has the potential to predict various traits relevant for animal breeding and herd management [14,15], including feed efficiency and energy intake [16], ketone bodies [17], lameness [18], fertility [19] and pregnancy [20,21].

Some studies, applying diverse methods, have demonstrated the potential of MIR spectral data to predict mastitis [22,23,24]. For instance, one study [22] used MIR spectra of cow milk to estimate the content of lactoferrin (LTF), a glycoprotein which is positively associated with immune response. In a next step, MIR predicted LTF and a somatic cell count-derived score (SCS) were used to develop models to predict presence of mastitis. External validation showed high sensitivities of 94.9% (SCS), 94.7% (SCS + LTF), and 95.4% (SCS + LTF + SCS × LTF) but low specificities of 34.4%, 37.6%, and 32.3%. Another study aimed to develop an MIR-based early warning system for mastitis [23]. In that study, test-day data were linked to routinely collected mastitis diagnosis data. Test-day records within 7 days before diagnosis and with SCC > 400,000 were defined as mastitis cases; records not linked to a mastitis diagnosis and with SCC < 50,000 were defined as healthy cases. The MIR-based model showed sensitivity of 75.6% and specificity of 83.3%. In a third study [24], models to predict mastitis were developed based on MIR spectra alone, SCS alone, or the two predictors together. For model calibration, test-day records within 7 days of diagnosis (before or after) were considered as mastitis cases, whereas in validation the period was defined as 21 days between test-day and date of diagnosis. The models showed specificities of 70.8% if based on MIR alone, 84.9% if based on SCS alone, and 79.1% if based on both predictors. The corresponding sensitivities were 53.4%, 50.1% and 57.4%.

The objective of the present study was to develop and evaluate prediction models based on MIR spectral data and SCS from routine milk recording to predict bovine clinical mastitis, diagnosed by herd veterinarians; and to explore whether there are differences in their prediction accuracies for acute and chronic mastitis diagnoses. An additional aim was to examine predicted mastitis probabilities of those models in time periods before and after mastitis diagnosis.

## 2. Materials and Methods

### 2.1. Data and Data Preparation

Data for this study were collected between July 2014 and January 2020 within the Austrian routine milk recording system and its health monitoring system (GMON) [4]. Data included test-day milk records, MIR spectral analysis data of milk, and clinical diagnoses of mastitis by veterinarians of 59,002 dairy cows of Fleckvieh (dual purpose Simmental), Holstein Friesian and Brown Swiss breeds from 2621 farms. We included only data from validated GMON farms for which at least 75% of mastitis diagnoses were uploaded electronically by the veterinarians themselves, instead of reported by the farmers to milk inspectors. The GMON diagnostic data are recorded by veterinarians using a coding system, which only includes diseases which can be diagnosed on site. For the present study, clinical mastitis diagnoses, subdivided into acute mastitis and chronic mastitis, were used. The distinction between acute and chronic mastitis, required by GMON, was done by the herd veterinarian based on clinical examination and lactation history of the animal [4,25].

In Austria, milk testing is performed 9–11 times a year, which translates to once every 33–41 days. The following data were provided for each test-day: anonymized herd and cow ID, date of test-day, breed, parity, days in milk (DIM), milk yield, fat, protein, urea, and somatic cell count (SCC) and MIR spectral data. Only test-day records from lactation days 3 to 365 were considered. Given the relatively small numbers of records from cows with at least five 5 or more lactations, these records were grouped into one parity category of ‘parity 5+’.

Data were merged and initially processed using SAS (SAS Institute Inc., Cary, NC, USA); see Table 1. Test-day records within 21 days before and 21 days after mastitis diagnosis were defined as mastitis cases. Cows were considered healthy if their test-day records fell outside 21 days before or after diagnosis, or if no diagnosis was recorded for them. The interval of 21 days between test-day and diagnosis enabled detection of almost all potential mastitis events between two test-days, and it ensured compatibility of the prediction method with the Austrian milk recording system.

Further data processing and analyses were performed within R software [26]. Daily means (including 95% confidence intervals) of test-day SCC were plotted for 21 days before and after diagnoses of acute or chronic mastitis separately (Figure 1). A density plot was produced to visualize the distribution of test-day records linked to acute and chronic mastitis diagnoses during 365 days of lactation (Figure 2a). Frequencies of acute and chronic diagnoses were plotted for the parity classes 1, 2, 3, 4, and 5+ (Figure 2b).

### 2.2. Pre-Treatment of MIR Spectra

Milk samples were analyzed in official laboratories of the Austrian milk recording system using spectrometers (FOSS^®^ Instruments, Hillerød, Denmark). Resulting MIR spectra express the absorbance of infrared light at 1060 wavenumbers from 926 to 5010 cm^−1^ [27]. Spectral data from different labs were routinely standardized in order to make spectra comparable across time and instruments [28]. To maximize useful information and reduce noise, 212 data points were collected from the following ranges: 968.1 to 1577.5 cm^−1^; 1731.8 to 1762.6 cm^−1^; 1781.9 to 1808.9 cm^−1^ and 2831 to 2966 cm^−1^ [29]. First derivatives of the 212 spectral data points were calculated using the Savitzky–Golay filter and a width of 5, in accordance with relevant studies [10,11,18,21,23,24]. The first derivatives were corrected for DIM according to Vanlierde et al. [30]: each value was multiplied by (1) a constant (i.e., 1), (2) a linear term (√3 × x) and (3) a quadratic [√5/4 × (3x^2^ − 1)] modified Legendre polynomial [31], where
x = −1 + 2[(DIM − 3)/(305 − 3)].

These steps generated 212 constants, 212 linear terms and 212 quadratic terms as spectral variables to be used as predictors in the models.

To achieve an approximately normal distribution of SCC, counts were logarithmically transformed into an SCS using the following formula [32]:SCS = log2 (SCC/100,000) + 3. 

SCS was then used as a variable in the prediction models.

### 2.3. Prediction Models

Models were developed based on MIR spectral data alone, SCS alone, or both. Fixed effects in all models were parity (1, 2, 3, 4, 5+) and breed (Fleckvieh, Holstein Friesian, Brown Swiss). Mastitis cases were predicted using partial least squares discriminant analysis (PLS-DA), and models were tuned using the function ‘trainControl’ in the R package caret [33], with the following parameters: 10-fold cross validation (for fine tuning); automatic setting of the number of components, with the maximum set at 60 to avoid overfitting; centering and scaling of spectral values; and discrimination by class probabilities, with a cut-off at 0.5.

The complete dataset of 1,183,282 records was randomly split into a calibration dataset (~70%) and validation dataset (~30%) based on the farm (*n* = 2621). Sampling was performed without replacement using the function ‘sample’ in the R package base [26]. This approach allowed models to be validated externally, i.e., against data of farms, being independent from those used to calibrate the models. Random down sampling was used to balance healthy and mastitis cases in calibration, because only approximately 1.7% of records in the dataset were linked to mastitis diagnoses. No such balancing was performed on the validation dataset. During model calibration, only healthy and mastitis cases were distinguished, not acute and chronic cases of mastitis.

Model performance was quantified in terms of the following: sensitivity, defined as the proportion of mastitis cases correctly classified; specificity, defined as the proportion of healthy cases correctly classified; and balanced accuracy, defined as the mean of sensitivity and specificity. Model performance was also assessed using receiver operating characteristic (ROC) curves, which plot the true positive rate (sensitivity) against the false positive rate (1—sensitivity), and the resulting area under those curves (AUC). AUC ranges from 0.5 (no discrimination) to 1.0 (perfect discrimination) [34,35]. Performance indicators were measured against the validation dataset in 10 independent replicates.

To explore whether model accuracy depended on the interval between test-day and mastitis diagnosis, we analyzed diagnostic performance in six time windows before mastitis diagnosis (indicated with a negative sign) and after mastitis diagnosis (indicated with a positive sign) [24]. Data splitting into time windows affected only mastitis cases, not healthy cases, so diagnostic specificity did not vary across the six time windows. Table 2 displays the average numbers of healthy and mastitis records in the calibration dataset after random down sampling, and the numbers of records in the validation dataset overall and across the six time windows.

In order to investigate model performance separately for acute and chronic mastitis diagnoses, sensitivity, balanced accuracy and AUC were evaluated separately for the two types of clinical mastitis diagnoses in the validation dataset, which did not affect specificity. Performance differences were assessed for significance using pairwise *t* tests. Furthermore, sensitivity and balanced accuracy were plotted for the six time windows before and after mastitis diagnosis. Additionally, probabilities of mastitis derived from prediction models based on MIR spectral data alone, SCS alone or both were plotted for test-day records defined as mastitis cases separately for acute and chronic mastitis events. Separate prediction models were not developed for acute and chronic cases, since this was not the objective of this study.

## 3. Results

### 3.1. Descriptive Statistics

Analysis of daily mean SCC at 21 days before and after mastitis diagnosis (Figure 1) showed that in animals with acute mastitis, SCC increased steadily before diagnosis, especially around 7 days before diagnosis. After diagnosis, SCC rapidly decreased until day 3. In animals with chronic mastitis, SCC increased leading up to mastitis diagnosis, although the trend was weaker and the range of variation larger.

Frequency of clinical mastitis diagnosis, especially acute cases, peaked during the first 50 days of lactation (Figure 2a). Frequency of chronic disease peaked again during the last 90 days of the 365-day lactation period. Frequency of acute cases increased with parity from 1% for parity 1 to 2% for parity 5+ (Figure 2b). Frequency of chronic cases was generally lower than that of acute cases, and frequency was only slightly higher for parity 3, 4 and 5+ than for parity 1 and 2.

### 3.2. Prediction of Mastitis Based on SCS and MIR Spectral Data

PLS-DA during model calibration (Table 3) showed that the model based only on MIR led to significantly higher sensitivity than the model based only on SCS (0.620 vs. 0.610), but significantly lower specificity (0.697 vs. 0.725). The model based on both MIR and SCS showed significantly higher sensitivity (0.657) and specificity (0.763) than the other two models. The combined model also showed significantly higher balanced accuracy (0.698) than the models based only on MIR (0.645) or SCS (0.666). Similar results were obtained with the validation dataset. AUC, only calculated for the validation dataset, showed a higher value for the model based on MIR and SCS (0.760) than the models based only on MIR (0.696) or SCS (0.722).

When prediction accuracy was assessed during different time windows, all three models performed better closer to mastitis diagnosis (day 0) (Table 4). All models showed the highest sensitivity, balanced accuracy and AUC in the time window ‘0 to +7 days’ after diagnosis, when AUC was significantly higher for the model based on MIR and SCS (0.849) than for the models based only on MIR (0.767) or SCS (0.828). The respective values were lower in the corresponding time window of ‘−7 to 0 days’ before diagnosis: 0.787, 0.708, and 0.770. In every time window, balanced accuracy and AUC were significantly higher for the model based on MIR and SCS than for models based on MIR alone or SCS alone.

Sensitivity of the model based only on SCS was significantly better than the model based only on MIR for the time windows ‘−14 to −8 days’, ‘−7 to 0 days’ and ‘0 to +7 days’. Conversely, sensitivity of the MIR model was significantly higher than that of the SCS model for the time windows ‘+8 to +14 days’ and ‘+15 to +21′ days. During the time window ‘−21 to −15 days’, sensitivity was similar for the models based on SCS only and based on MIR only, and it was significantly higher for the model based on MIR and SCS.

### 3.3. Prediction of Acute and Chronic Cases

The models based on MIR alone or both MIR and SCS were significantly better at predicting acute mastitis than chronic mastitis, based on sensitivity, balanced accuracy and AUC. The same was true for the model based on SCS alone, except that AUC did not differ significantly between the two types of diagnosis. AUCs for acute or chronic mastitis were 0.699 and 0.690 for the model based only on MIR, 0.723 and 0.719 for the model based only on SCS, and 0.762 and 0.756 for the model based on both. Comparison of model sensitivity and balanced accuracy between acute and chronic mastitis (Figure 3) showed that the model based on MIR performed significantly differently only in the time window ‘0 to −7 days’, when it predicted acute cases better. The model based only on SCS predicted chronic mastitis significantly better than acute mastitis in the time windows ‘−21 to −15 days’ and ‘−14 to −8 days’, whereas it predicted acute disease significantly better in the time windows ‘−7 to 0 days’, ‘0 to +7 days’ and ‘+8 to +14 days’. The model based on both MIR and SCS showed similar results as the model based on SCS, but the differences between acute and chronic mastitis achieved significance only in the time windows ‘−7 to 0 days’ and ‘0 to +7 days’.

### 3.4. Predicted Probabilities of Acute and Chronic Mastitis

The average predicted probabilities for records defined as mastitis cases were significantly different for the three models: 0.565 ± 0.055 (MIR), 0.5740 ± 0.058 (SCS) and 0.584 ± 0.066 (MIR + SCS). Tracking of the predicted probabilities of mastitis during 21 days before and after diagnosis (Figure 4) showed that the model based on MIR alone gave higher average probabilities for acute mastitis after diagnosis than before it (Figure 4a), whereas there was no clear trend in the differences of the probability of chronic disease before and after diagnosis (Figure 4b). The model based on SCS only showed a strong peak in probability for records within 1 day of acute mastitis (Figure 4c), and a slight trend toward higher probabilities for records closer to diagnosis with chronic mastitis (Figure 4d). The model based on MIR and SCS gave higher overall probabilities than the two other models. It showed a strong peak for records quite close to diagnosis of acute mastitis (Figure 4e), but no clear trend in probabilities of diagnosis of chronic disease (Figure 4f). For all three models, there was a higher variability in the probabilities for chronic cases compared to acute cases.

## 4. Discussion

In this study we explored the performance of models based on MIR spectral data and SCS to predict clinical mastitis of dairy cows within the setting of routine milk recording in Austria. Hence, study design was observational using routinely collected milk recording data and clinical mastitis diagnosis data of GMON. Since the distinction between acute and chronic mastitis was made according to the clinical judgement of the respective herd veterinarian, standardization of diagnosis data may have limitations, compared to an experimental study setting [25]. Nonetheless, data analyzed in this study represent the practical real-world situation on farms and the results can therefore be used in making decisions on-farm as well as in breeding.

Analysis of daily mean SCC during 21 days prior mastitis diagnosis (Figure 1) indicates that SCC could provide an early warning signal for farmers. However, changes in SCC before clinical disease, depending on the type of pathogen, were reported. For example, SCC can remain elevated for long periods before clinical signs of mastitis caused by *Staphylococcus aureus* [36]. Conversely, mastitis caused by *Escherichia coli* results in relatively low SCC before clinical occurrence, and falling rapidly thereafter. In addition, antibiotic treatment may lead to a rapid decrease in SCC after mastitis diagnosis, especially with acute cases. These factors should be considered for the interpretation of SCC in association with clinical mastitis, and may limit the reliability of SCC as an early indicator of clinical mastitis, since test-days are usually 4–5 weeks apart in milk recording schemes. Unfortunately, data on bacteriological testing for specific pathogens were unavailable in the present study, as not routinely recorded.

The peak of test-day records linked to clinical mastitis diagnosis during the first 50 days of lactation (Figure 2a) is consistent with the fact that clinical mastitis occurs most frequently in early stage of lactation [36,37,38]. The peak of chronic mastitis diagnoses in the last quarter of lactation may simply result from fact that chronic mastitis is characterized by recurrence.

The finding that frequency of acute mastitis diagnoses increased with parity number (Figure 2b), is consistent with other studies [37,38] reporting a higher risk of mastitis in multiparous cows.

The results of this study showed that the model based SCS and MIR, compared to models with SCS or MIR alone, performed best in terms of all the parameters (Table 3). This is consistent with a previous study on a smaller dataset [24]. That study found that models based on MIR alone, SCS alone or both MIR and SCS gave substantially higher specificities (0.708–0.849) than sensitivities (0.501–0.574). The models in the present study gave more balanced values of specificity and sensitivity, which may reflect the same definition of mastitis in the calibration and validation datasets (an interval between test-day and diagnosis within 21 days). Another reason may be that we used a different modelling function and included cross-validation and automatic determination of the number of components for fine tuning of the model.

The balanced accuracies in the present study are higher than those reported for models based on SCS and MIR predicted LTF (0.639–0.662) [22]. This is interesting because that previous work defined mastitis cases more narrowly based on records within 7 days of the mastitis event.

Based on the definitions of Simundic [34], the AUCs in Table 3 indicate sufficient diagnostic accuracy for the model based only on MIR and good accuracy for the models based only on SCS or on both MIR and SCS. These results suggest that SCS performs better than MIR as a single predictor of mastitis, but that combining the two factors gives better results.

The investigation of test-day records in different time windows before and after mastitis diagnosis clearly indicated higher prediction accuracy for test days close to diagnosis. It also confirmed the advantage of combining MIR spectral data with SCS. In general, the SCS model performed better than the MIR model, but in two time windows post diagnosis, the MIR model showed higher prediction accuracies compared to SCS. This may be due to antibiotic treatments that dampen immune reactions and reduce the number of somatic cells. Regarding AUC, it is difficult to compare results of the present study with the AUC of 0.83 reported by Dale and Werner [23], because those investigators included healthy records only if SCC < 50,000 records and mastitis records only if SCC > 400,000.

All three models in the present study predicted acute mastitis better than chronic mastitis across the entire time window ‘−21 to +21 days’. However, prediction accuracy depended on the interval between test-day and mastitis diagnosis. Shortly after mastitis diagnosis (‘0 to +7 days’), all models predicted acute cases better than chronic cases; but in other time windows before diagnosis, the models based only on SCS or on both MIR and SCS predicted chronic cases better than acute cases. Thus, although the models generally were better at predicting acute mastitis, they should be useful for predicting both types of disease, increasing their value for farm management and genetic evaluation of the trait udder health. To this end, our finding that prediction accuracy was related to probability (Figure 4) opens up the possibility of creating mastitis risk classes based on probability that may be useful for monitoring udder health on dairy farms.

The methodology presented in the current study lends itself to application with more frequent recording of SCS and MIR or other spectral data, e.g., in farms with automatic milking systems [39]. Furthermore, studies that include a large number of bacteriological tests and that identify types of pathogens could provide deeper insights into the prediction of mastitis, also for subclinical mastitis cases, using MIR spectral data and SCS.

## 5. Conclusions

The results of this study indicate that mastitis events may be predicted somewhat better by including information from MIR spectral data compared to using SCS alone. We suggest the combined use of SCS and MIR spectral data for the development of a mastitis screening tool, indicating risk classes, for farmers as part of routine milk recording. Such a screening tool cannot replace veterinary diagnosis via animal examination, but can aid the farmer in monitoring udder health. For genetic evaluation of udder health, it may be useful to add MIR-based mastitis probabilities to the ‘udder health index’, with current relative weights of 70% for SCS and 30% for clinical mastitis diagnosis from GMON; diagnosis data are only partially available from Austrian dairy farms, compared to SCS and MIR spectra which are routinely available for almost all farms. For that purpose, an ongoing study investigates heritabilities and genetic correlations of SCS, clinical mastitis diagnosis and MIR-predicted mastitis probabilities.

## Figures and Tables

**Figure 1 animals-12-01830-f001:**
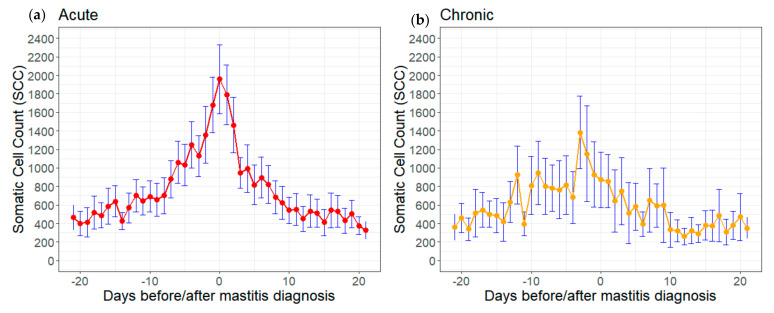
Daily trend of means and 95% confidence intervals of somatic cell count (SCC) in the time period 21 days before and after diagnosis of (**a**) acute or (**b**) chronic clinical mastitis diagnoses.

**Figure 2 animals-12-01830-f002:**
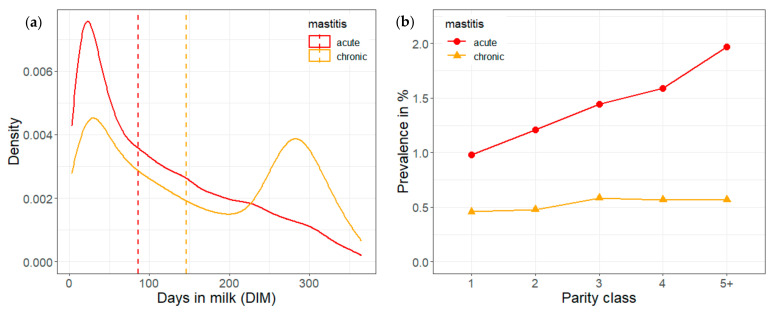
Distribution of test-day records linked to acute and chronic clinical mastitis diagnoses during 365 days of lactation. (**a**) Density as a function of days in milk, vertical lines indicating median values. (**b**) Incidence as a function of parity.

**Figure 3 animals-12-01830-f003:**
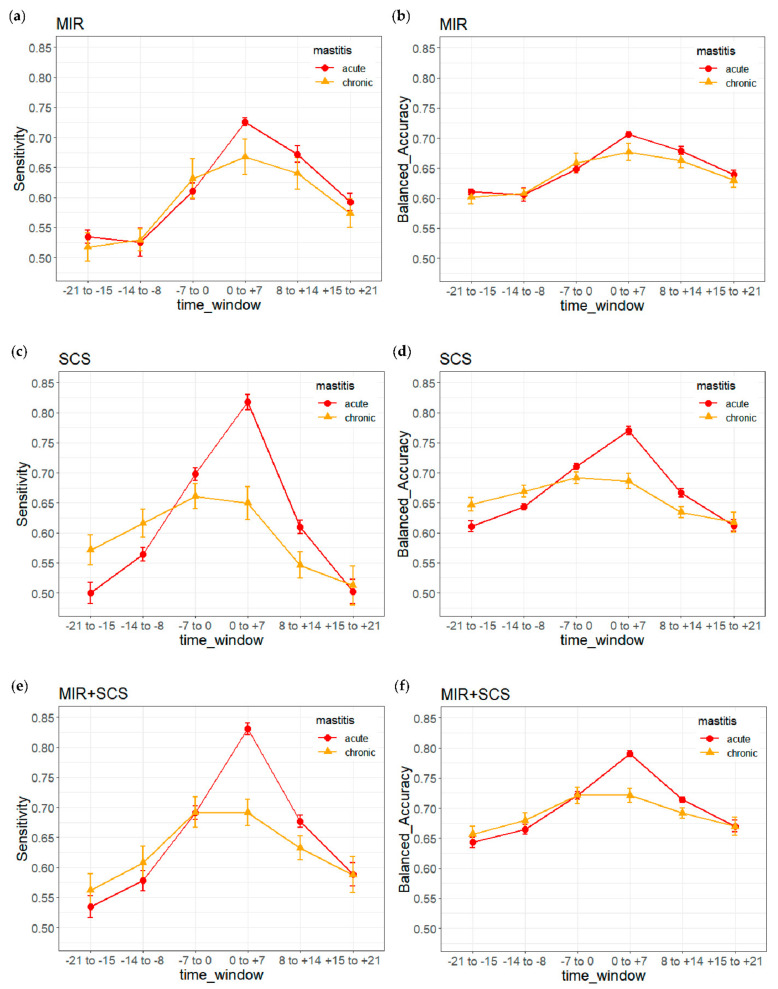
Sensitivities and balanced accuracies for models based on different predictor variables (MIR, SCS, MIR + SCS) against the validation dataset as a function of time windows before or after mastitis diagnosis: sensitivities of MIR (**a**), balanced accuracies of MIR (**b**), sensitivities of SCS (**c**), balanced accuracies of SCS (**d**), sensitivities of MIR+SCS (**e**), balanced accuracies of MIR+SCS (**f**). Means and 95% confidence intervals from 10 independent runs are shown.

**Figure 4 animals-12-01830-f004:**
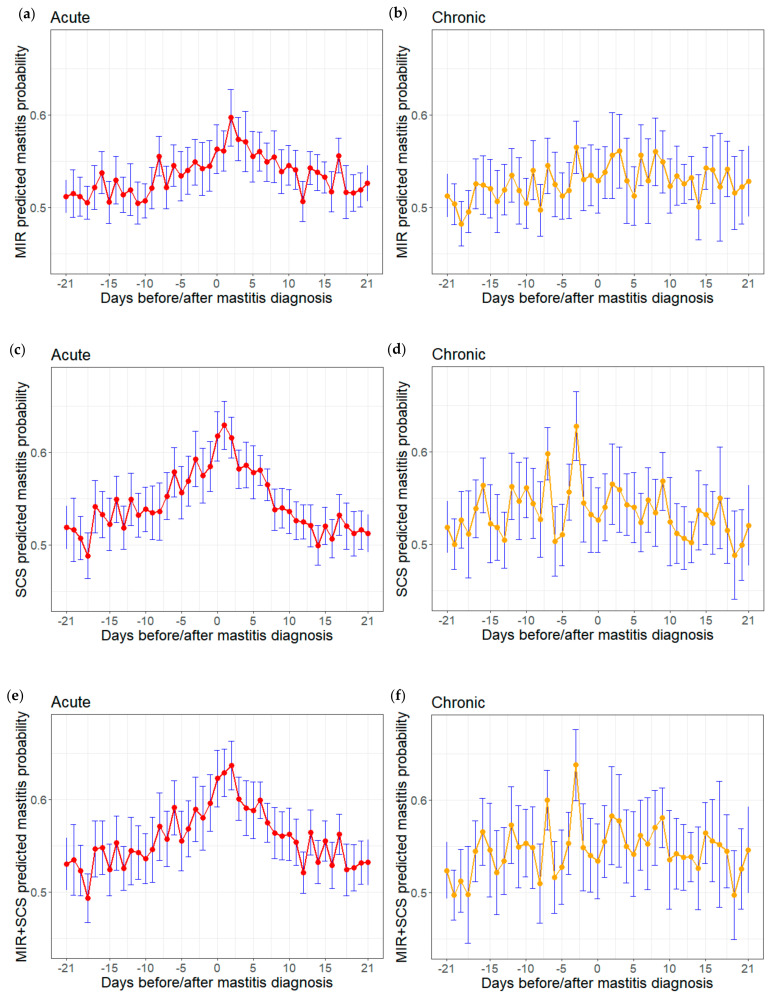
Predicted probabilities of acute (**left**) or chronic (**right**) mastitis for test-day records linked to diagnosis within 21 days. Predictions were generated using models based on MIR only, SCS only or both: MIR acute (**a**), MIR chronic (**b**), SCS acute (**c**), SCS chronic (**d**), MIR+SCS (**e**) and MIR+SCS chronic (**f**). Daily means with 95% confidence intervals are shown.

**Table 1 animals-12-01830-t001:** Properties of the dataset used for analysis.

Unit	Records (n)
Farms	2621
Cows	59,002
Fleckvieh (Dual purpose Simmental)	46,042
Holstein Friesian	7645
Brown Swiss	5315
Test-day records	764,542
Mastitis records ^1^	12,656
Acute	8917
Chronic	3739

^1^ test-day records within 21 days before or after mastitis diagnosis.

**Table 2 animals-12-01830-t002:** Numbers of healthy and mastitis records in the calibration dataset (after random down sampling) and validation dataset (for different time windows).

Dataset	Time Window ^1^ (days)	Records (*n*) *	
		Healthy	Mastitis
Calibration	−21 to +21 (overall)	8846	8846
Validation	−21 to +21 (overall)	224,202	3810
	−21 to −15	224,202	586
	−14 to −8	224,202	629
	−7 to 0	224,202	738
	0 to +7	224,202	651
	+8 to +14	224,202	647
	+15 to +21	224,202	643

* Means of 10 runs. ^1^ days of test-day date before or after mastitis diagnosis.

**Table 3 animals-12-01830-t003:** Model performance against calibration and validation datasets.

Dataset	Predictor Variables	Sensitivity	Specificity	Balanced Accuracy	AUC
calibration	MIR	0.620 ^a^ (0.005)	0.697 ^a^ (0.005)	0.658 ^a^ (0.004)	-
	SCS	0.610 ^b^ (0.003)	0.725 ^b^ (0.005)	0.668 ^b^ (0.003)	-
	MIR + SCS	0.657 ^c^ (0.003)	0.763 ^c^ (0.005)	0.710 ^c^ (0.003)	-
validation	MIR	0.605 ^a^ (0.010)	0.686 ^a^ (0.005)	0.645 ^a^ (0.004)	0.696 ^a^ (0.005)
	SCS	0.610 ^a^ (0.010)	0.722 ^b^ (0.008)	0.666 ^b^ (0.004)	0.722 ^b^ (0.004)
	MIR + SCS	0.645 ^b^ (0.007)	0.751 ^c^ (0.006)	0.698 ^c^ (0.003)	0.760 ^c^ (0.005)

AUC = area under the receiver operating characteristic curve. MIR = first derivatives of 212 selected MIR spectral variables corrected for days in milk. SCS = somatic cell score. Results are the mean (SD) of 10 independent runs. Values with different superscripts (a, b, c) in the same column within the calibration or validation dataset differ significantly (Bonferroni–Holm method, *p* < 0.05).

**Table 4 animals-12-01830-t004:** Sensitivity, balanced accuracy and AUC (means of 10 independent runs, SD in parentheses) of different predictor variables in validation, split up for individual time windows before or after mastitis diagnosis.

Time Window	Predictor Variables	Sensitivity	Balanced Accuracy	AUC
−21 to −15	MIR	0.529 ^a^ (0.017)	0.607 ^a^ (0.007)	0.644 ^a^ (0.009)
	SCS	0.524 ^a^ (0.018)	0.623 ^b^ (0.008)	0.660 ^b^ (0.008)
	MIR + SCS	0.544 ^b^ (0.015)	0.647 ^c^ (0.006)	0.696 ^c^ (0.008)
−14 to −8	MIR	0.526 ^a^ (0.022)	0.606 ^a^ (0.010)	0.650 ^a^ (0.010)
	SCS	0.581 ^b^ (0.017)	0.652 ^b^ (0.008)	0.695 ^b^ (0.012)
	MIR + SCS	0.587 ^b^ (0.017)	0.669 ^c^ (0.006)	0.722 ^c^ (0.009)
−7 to 0	MIR	0.617 ^a^ (0.021)	0.651 ^a^ (0.010)	0.708 ^a^ (0.012)
	SCS	0.686 ^b^ (0.013)	0.704 ^b^ (0.006)	0.770 ^b^ (0.007)
	MIR + SCS	0.691 ^b^ (0.015)	0.721 ^c^ (0.009)	0.787 ^c^ (0.011)
0 to +7	MIR	0.709 ^a^ (0.014)	0.697 ^a^ (0.007)	0.767 ^a^ (0.009)
	SCS	0.769 ^b^ (0.021)	0.746 ^b^ (0.010)	0.828 ^b^ (0.012)
	MIR + SCS	0.790 ^c^ (0.014)	0.771 ^c^ (0.007)	0.849 ^c^ (0.008)
+8 to +14	MIR	0.664 ^a^ (0.014)	0.675 ^a^ (0.007)	0.727 ^a^ (0.010)
	SCS	0.594 ^b^ (0.012)	0.650 ^b^ (0.007)	0.714 ^b^ (0.005)
	MIR + SCS	0.666 ^a^ (0.015)	0.708 ^c^ (0.006)	0.772 ^c^ (0.007)
+15 to +21	MIR	0.587 ^a^ (0.019)	0.637 ^a^ (0.010)	0.681 ^a^ (0.011)
	SCS	0.505 ^b^ (0.028)	0.614 ^b^ (0.014)	0.662 ^b^ (0.014)
	MIR + SCS	0.588 ^a^ (0.026)	0.670 ^c^ (0.013)	0.732 ^c^ (0.015)

Time window = days of test-day date before or after mastitis diagnosis. AUC = area under the receiver operating characteristic curve. MIR = first derivatives of 212 selected MIR spectral variables corrected for days in milk. SCS = somatic cell score. Values with different superscripts (a, b, c) in the same column within the calibration or validation dataset differ significantly (Bonferroni–Holm method, *p* < 0.05).

## Data Availability

The data presented in this study are available on request from the corresponding author. The data are not publicly available due to privacy restrictions of the data provider and owner, the Austrian milk recording system (LKV Austria Gemeinnützige GmbH).

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
