# Peer review of "Prediction of Acute and Chronic Mastitis in Dairy Cows Based on Somatic Cell Score and Mid-Infrared Spectroscopy of Milk"

_animals, 2022, doi:10.3390/ani12141830_

Round 1

Reviewer 1 Report

The reviewed manuscript presents an interesting idea, but editing is required to strengthen the scientific soundness of the results as discussed. 

Line 27-28: Split into 2 separate sentences

Line 62: should be start of a new paragraph

Line 67-68: repetitive from previous paragraph

Line 75-76: rewrite for clarity

Line 79: what does this “test-day records within 7 days before diagnosis” actually mean in the context of diagnosis. What is being evaluated? As read, it just sounds like the existence of test-day records results in a diagnosis. Same as in lines 84-85

Line 90: While I understand the premise of the paper, I think authors should re-evaluate discussion of clinical mastitis diagnoses as being based on SCS (or SCC). Clinical mastitis, by definition, would be identifiable by abnormalities in the milk or udder. Of course, the SCC would be elevated as the clinical signs are a result of these changes. This needs to be address throughout the manuscript in multiple different locations.

Introduction lacks flow, reads very “choppy”

Line 108 How were acute vs chronic mastitis cases ACTUALLY diagnosed by the veterinarians. What about subclinical vs clinical?

Line 108: Were animals treated with antibiotics? This would impact data tremendously. Other incidence of disease such as transition cow diseases, etc.?

Line 280: Address parenthesis

Line 306: See comment about clinical mastitis (from line 90)

Line 332-335: This needs further explanation, especially since it was not discussed in data collection in the Materials and Methods. If farmers are submitting these to determine dry off therapies, how are they coded as chronic? Without additional information in the manuscript, I would classify these as subclinical rather than chronic, though they could be both and in the current collection scheme, there is no way to determine. Again, as the manuscript is currently written with information provided.

Line 355: Edit for proper grammar

Much of the discussion simply repeats results. Please correct

Line 390-392: This takes findings a bit far, especially with flaws in manuscript and reporting as it currently stands.

Line 392-394: Also a reach based on data reporting as it currently stands.

Author Response

Thank you for your valuable review of our manuscript. We've triet to meet all points in the revision as best as possible.

Author Response

Thank you for your valuable review of our manuscript. We've tried to meet all points in the revision as best as possible.

Reviewer 3 Report

Additional remarks:

The quality and the scientific significance of this manuscript are good, but I find only some problematic parts.

Detailed review:

lines 107-108: How could veterinarians decide between acute and chronic mastitis? What are the threshold limits?

Figure 1: please reedit this Figure, put a) and b) letters same page as body of Figure!

lines 280-281: please delete bolded sentence!

Figure 3: please cite this Figure in the main text!

lien 332: early stage of lactation instead of early lactation!

lines 392-396: must be rewrite the conclusion this section! Please modify these sentences based on own results!

Author Response

(The authors gave the same response as above.)

Round 2

Reviewer 1 Report

The authors are thanked for diligently addressing all comments and for providing clarity on points of confusion. The dedication to providing important data in this manuscript is evident. The authors have tremendously improved many facets of the manuscript, thus minor changes are suggested. 

Response to Point 5: The explanation is tremendously informative. Perhaps a slight edit would further enhance clarity. “In that study, test-day records linked to mastitis routinely collected mastitis diagnosis data within 7 days before diagnosis and with SCC >400,000 were defined as mastitis cases…”

Response to Point 6: Further explanation provided by the authors is valued. It is possible that the confusion or disagreement could be ameliorated with a slight change similar to this: “The objective of the present study was to develop and evaluate prediction models 94 based on MIR spectral data and SCS from routine milk recording to predict clinical mastitis diagnoses of dairy cows in advance of disease detection by trained veterinarians,”

Response to Point 7: See additional comments for improved flow

Response to Point 8: Addition of the included paragraph is a tremendous improvement. Consider moving paragraph towards the end of the discussion. Or perhaps included, “Nonetheless” at the beginning of the sentence on Line 320.

Response to Point 9: Consider adding back in deleted sentence on line 338-339 “may limit the reliability of SCC as an early indicator of clinical mastitis, since test-days 338 are usually 4-5 weeks apart in milk recording schemes” to augment the inclusion of lines 332-338

Response to Point 10: Apologies. The parenthesis reads “(Error! Reference source not found)”

Additional comments

Lines 60-62: Combine with previous paragraph

Lines 63-67: Combine with previous paragraph

Author Response

Dear Reviewer, thank you again for your comments and suggestions to further improve our manuscript. We followed all points.
